# Model estimations of geophysical variability between satellite measurements of ozone profiles

Patrick E. Sheese[1], Kaley A. Walker[1], Chris D. Boone[2], Doug A. Degenstein[3], Felicia Kolonjari[4], David Plummer[5], Douglas E. Kinnison[6], Patrick Jöckel[7], and Thomas von Clarmann[8]

[1]University of Toronto, Department of Physics, Toronto, Canada
[2]University of Waterloo, Department of Chemistry, Waterloo, Canada
[3]University of Saskatchewan, ISAS, Department of Physics and Engineering, Saskatoon, Canada
[4]Environment and Climate Change Canada, Victoria, Canada
[5]Environment and Climate Change Canada, Climate Research Branch, Montreal, Canada
[6]National Center for Atmospheric Research, Atmospheric Chemistry Observations & Modeling Laboratory, Boulder, USA
[7]Deutsches Zentrum für Luft- und Raumfahrt (DLR), Institut für Physik der Atmosphäre, Oberpfaffenhofen, Germany
[8]Karlsruhe Institute of Technology, Institute of Meteorology and Climate Research, Karlsruhe, Germany

*Correspondence*: Kaley A. Walker (kaley.walker@utoronto.ca)

**Abstract.** In order to validate satellite measurements of atmospheric composition, it is necessary to understand the range of random and systematic uncertainties inherent in the measurements. On occasions where measurements from two different satellite instruments do not agree within those estimated uncertainties, a common explanation is that the difference can be assigned to geophysical variability, i.e. differences due to sampling the atmosphere at different times and locations. However, the expected geophysical variability is often left ambiguous and rarely quantified. This paper describes a case study where the geophysical variability of $O_3$ between two satellite instruments, ACE-FTS (Atmospheric Chemistry Experiment – Fourier Transform Spectrometer) and OSIRIS (Optical Spectrograph and InfraRed Imaging System), is estimated using simulations from climate models. This is done by sampling the models CMAM (Canadian Middle Atmosphere Model), EMAC (ECHAM/MESSy Atmospheric Chemistry), and WACCM (Whole Atmosphere Community Climate Model) throughout the upper troposphere and stratosphere at times and geolocations of coincident ACE-FTS and OSIRIS measurements. Ensemble mean values show that in the lower stratosphere $O_3$ geophysical variability tends to be independent of the chosen time coincidence criterion, up to within 12 h; and conversely, in the upper stratosphere geophysical variation tends to be independent of the chosen distance criterion, up to within 2000 km. It was also found that in the lower stratosphere, at altitudes where there is the greatest difference between air composition inside and outside the polar vortex, the geophysical variability in the Southern polar region can be double of that in the Northern polar region. This study shows that the ensemble mean estimates of geophysical variation can be used when comparing data from two satellite instruments to optimize the coincidence criteria, allowing for the use of more coincident profiles while providing an estimate of the geophysical variation within the comparison results.

## 1 Introduction

A significant uncertainty when comparing concentrations of trace species measured from different satellite instruments is the difference due to the satellites sampling the atmosphere at different times and locations ("coincident" measurements are never truly coincident). This uncertainty can be called "geophysical variability", "natural variability", or "coincident location uncertainty"—this study uses the term geophysical variability. Loew et al. (2017), when reviewing the methods and techniques used in Earth Observation data validation, wrote "Collocated measurements should be close to each other relative to the spatiotemporal scale on which the variability of the geophysical field becomes comparable to the measurement uncertainties," and it is assumed that the "spatiotemporal scale" (coincidence criteria) that will result in geophysical variability on the order of the measurement uncertainties is known. However, it is often the case that validation studies involving satellite-based atmospheric measurements will choose coincidence criteria without discussing the geophysical justification of the criteria.

There are many validation studies that try to either estimate or limit geophysical variability using various techniques. One common method for reducing temporal variability is to make use of chemical models in order to diurnally scale the measurements to a common local time (e.g., Sheese et al., 2016 and references therein). Two methods that are similar to each other are the trajectory mapping (Morris et al., 1995) and the target hunting techniques (Danilin et al., 2000) that involve tracking air parcels using forward and/or back trajectories when comparing two different data sets. These have been shown to be reliable tools for validation (e.g., Bacmeister et al., 1999; Morris et al., 2000; Danilin et al., 2002a; Danilin et al., 2002b; Liu et al., 2013) without introducing large sources of uncertainty, however it can be computationally expensive to create trajectories for multiple instrument data sets. Verhoelst et al. (2015) coupled a numerical weather forecast model with an ozone tracer model to create a high spatial resolution Observing System Simulation Experiment (OSSE) in order to model coincident mismatch uncertainty (as well as vertical smoothing uncertainty) between satellite- and ground-based measurements. Although it was shown that the OSSE could successfully represent the geophysical variability, as discussed by Loew et al. (2017), this method would likely not be suitable for atmospheric targets that exhibit greater geophysical variability than $O_3$. Simple statistical or chemistry models have also been used in studies to assess geophysical variability between atmospheric measurements (e.g., Aghedo et al., 2011; Guan et al., 2013; Toohey et al., 2013; Fassò et al., 2014; Sofieva et al., 2014; Millán et al., 2016).

In a similar, yet simplified, approach to Verhoelst et al. (2015), this study makes use of readily available output from three climate models that relaxed various meteorological fields using specified dynamics: the Canadian Middle Atmosphere Model (CMAM), the ECHAM/MESSy (European Centre Hamburg general circulation model, Modular Earth Sub-model System) Atmospheric Chemistry (EMAC) model, and the Whole Atmosphere Community Chemistry Model (WACCM). It is important to note that this study is not intended to validate either the ACE-FTS or OSIRIS $O_3$ data products. This is a case study that makes use of ACE-FTS and OSIRIS geolocation data and $O_3$ products to demonstrate how readily available data from nudged climate models can be used to estimate large scale geophysical variability between satellite measurements of atmospheric trace species, and how they can be used to make informed decisions when choosing coincidence criteria in a validation study. In

this study, given the horizontal resolution of the three climate models that were used, large scale variability is on the order of 200-300 km, which is on the order of the atmospheric path length of a limb viewing instrument at the tangent height.

The following section describes the satellite and model data sets used in this study, and Section 3 describes the methodology for sampling the model data and how the data sets are compared to one another. Section 4 discusses the resulting simulated geophysical variability and how those results can potentially be used to help improve validation studies. A summary is then given in Section 5.

## 2 Data descriptions

### 2.1 ACE-FTS on SCISAT

The ACE-FTS (Atmospheric Chemistry Experiment – Fourier Transform Spectrometer) instrument (Bernath et al., 2005) is a solar occultation instrument on board the Canadian satellite SCISAT, which was launched into a highly inclined, non-sun synchronous orbit in 2003. Since February 2004, ACE-FTS has been making observations of Earth's limb, providing profiles of atmospheric temperature and concentrations of over 30 trace species between altitudes of ~5 and 150 km. The instrument is a high-spectral resolution (0.02 cm$^{-1}$) infrared spectrometer detecting solar radiation between 750 and 4400 cm$^{-1}$.

The O$_3$ retrieval algorithm, described by Boone et al. (2005; 2013), is a global least-squares fitting technique that uses Levenberg-Marquardt iteration to converge on a solution without the need of a priori information. Version 3.5/3.6 data are used in this study, where the forward modelled spectra in 40 different microwindows between 829 and 2673 cm$^{-1}$ are calculated using spectral parameters from the HITRAN 2004 (Rothman et al., 2005) database with some updates, as described by Boone et al. (2013). Ozone is retrieved between 5 and 95 km assuming horizontal homogeneity, and CFC-12, HCFC-22, CFC-11, N$_2$O, CH$_4$, HCOOH, and H$_2$O, along with various isotopologues, are simultaneously retrieved as interfering species. The reported statistical fitting error, described by Boone et al. (2005; 2013), is typically on the order of 2-3% in the 10-15 km range and ~1.5-2% in the 15-55 km range. Dupuy et al. (2009) validated the ACE-FTS v2.2 ozone data set using correlative data from multiple satellite, ground-based, and balloon-based instruments, and Sheese et al. (2017) compared v3.5 O$_3$ data to correlative satellite data. In the upper troposphere to middle stratosphere, ACE-FTS v3.5 O$_3$ tends to exhibit a slight positive bias on the order of a few percent, and near 45-60 km, a positive bias on the order of 10-20%.

### 2.2 OSIRIS on Odin

The OSIRIS (Optical Spectrograph and InfraRed Imaging System) instrument (Llewellyn et al., 2004) is a limb scatter detector on board the Odin satellite, which was launched into a sun synchronous orbit in 2001 with a nominal ascending node of approximately 06:00 h local time. Since November 2001, OSIRIS has been observing Earth's limb, producing standard data products of O$_3$ and NO$_2$ profiles between altitudes of ~7 and 60 km, as well as various other atmospheric research products. The Optical Spectrograph is a grating spectrometer measuring between 275 and 810 nm with a spectral resolution of ~2 nm and a vertical field-of-view of ~1 km at the tangent point.

The O$_3$ retrieval algorithm is described by Bourassa et al. (2012) and uses a multiplicative algebraic reconstruction technique (Roth et al., 2007; Degenstein et al., 2009). Version 5.07 O$_3$ data are used in this study, where pressure and temperature profiles are obtained from the European Centre for Medium-Range Weather Forecasts (ECMWF), and ozone is retrieved in number density, taking into account UV and visible absorption, and NO$_2$ and aerosols are simultaneously retrieved as interfering species. The ECMWF pressure and temperature profiles are then used to convert the retrieved O$_3$ densities to volume mixing ratios. The reported OSIRIS O$_3$ uncertainties are typically on the order of 3-9% in the 10-55 km range.

Adams et al. (2013) found that the v5.07 OSIRIS data were in excellent agreement with coincident SAGE II profiles throughout the stratosphere, typically within 5%. Hubert et al. (2016) found there to be a statistically significant positive drift in the OSIRIS O$_3$ data above 20 km with respect to ozonesonde and lidar data. The OSIRIS drift is on the order of 1-3% dec$^{-1}$ between ~25 and 35 km, and increases to 8% dec$^{-1}$ near 42 km; however, this drift has been corrected in the v5.10 release (Bourassa et al., 2018).

## 2.3 Model data

Three different models were used in this study: CMAM, EMAC, and WACCM, all of which used specified dynamics to relax, or "nudge", different key atmospheric states (e.g. wind fields, temperature) to meteorological observations.

CMAM is a chemistry-climate model, described in detail by de Grandpré et al. (2000), Jonsson et al. (2004), and Scinocca et al. (2008). The CMAM30 simulation (McLandress et al., 2013), used in this study, is a 30-year run of the CMAM model with 6-hourly output from 1979 to 2010, on a 3.75° horizontal grid (linear T47 Gaussian grid). The model was run with 71 vertical levels up to 0.0007 hPa (~95 km) with vertical resolution on the order of 1 km around the tropopause, increasing to ~2.5 km in the mesosphere, and the dataset used here is comprised of six-hourly instantaneous model fields interpolated on to 63 constant pressure surfaces that span the full height range of the model. Below 1 hPa, temperatures and horizontal winds were nudged to 6-hourly values from ECMWF Interim Reanalysis (ERA-interim; Dee et al., 2011). CMAM simulations have been used in many studies to help understand stratospheric O$_3$ variations, climatology, and its effect on climate (e.g., Gillett et al., 2009, McLandress et al., 2011; Sakazaki et al., 2015; Froidevaux et al., 2019).

The global chemistry-climate model EMAC uses the general circulation model ECHAM version 5 as its base model in conjunction with MESSy version 2, which incorporates multiple sub-models, such as natural and anthropogenic emissions, land and ocean processes and interactions, and chemistry and transport (Jöckel et al., 2010; 2016). The simulations used in this study were on an approximate 2.8° horizontal grid (T42), with 90 vertical levels up to 0.01 hPa (~80 km). Within the 30-year run (1980-2010), the calculated divergence, vorticity, temperature, and logarithm of surface pressure variables were nudged above the boundary layer up to 10 hPa (with transition layers) to ERA-interim data with nudging times between 6 and 48 hours, depending on the variable. The data used in this study were from simulation RC1SD-base-10 (no nudging of global mean temperature), output every 5 hours (Jöckel et al., 2016). Multiple studies focusing on O$_3$ variations in the troposphere and stratosphere have used the EMAC model (e.g., Weber et al., 2011; Meul et al. 2014; Khosrawi et al., 2017).

WACCM is a climate chemistry model and is the atmospheric component of the National Center for Atmospheric Research's Community Earth System Model (Marsh et al., 2013). The simulations used in this study have horizontal resolutions of 1.9° latitude and 2.5° longitude, and have 88 vertical levels up to $5.1 \times 10^{-6}$ hPa (~140 km). The model simulation spans 1979 to 2013, and below 50 km the temperature, pressure, zonal and meridional wind, and surface stress variables were nudged to NASA's Modern Era Retrospective-Analysis for Research and Applications (MERRA) reanalysis data (Rienecker et al., 2011) with a 50-hour relaxation time constant. The WACCM model has been widely used to study $O_3$ variability throughout the atmosphere (e.g., Merkel et al., 2011; Brakebusch et al., 2013; Chandran et al., 2014).

Another set of WACCM simulations was used in this study, with the same setup, the only difference being that the output model data were directly output at the ACE-FTS and OSIRIS observation times and geolocations (individual observation profiles were assumed to be at a single time, latitude, and longitude, taken as the 30-km tangent height values). The WACCM output at the instrument Observed Locations will from here onward be referred to as WACCMOL.

All three models used in this study are considered to be "state-of-the-art" stratosphere-resolving chemistry climate models and regularly participate in multi-model intercomparisons, including the exhaustive model assessments performed for CCMVal-2 (SPARC CCMVal, 2010) and CCMI-1 (Morgenstern et al., 2017).

## 3 Methodology

In this study, altitude-dependent values of latitude and longitude were used for the measured profiles, however time values were assumed to be constant throughout a profile, taken as the mid-point of the measurement time. ACE-FTS and OSIRIS profiles were considered to be coincident if they were measured within 12 hours of each other and within 2000 km. In cases of multiple coincidences with a single profile, only the closest in latitude were chosen; hence each ACE-FTS profile has only one coincident OSIRIS profile and vice versa. Only data from 2004 to 2010 are used, as the latest start point out of all the data sets (model and instrument) was the ACE-FTS start of February 2004, and the CMAM and EMAC data sets both had the same earliest end point, December 2010.

In the following description the terms MOD and INST are used as general terms to indicate model and instrument values, respectively. The sampling of all three models (CMAM, EMAC, and WACCM) at satellite times and locations are done using the same methodology. First, for every instrument profile, the model $O_3$ data closest in time to $t_{INST}$ on both sides are isolated and are spline-interpolated in log-space from the native $(t_{MOD}, p_{MOD}, lon_{MOD}, lat_{MOD})$ grid to a $(t_{MOD}, z_{ACE}, lon_{MOD}, lat_{MOD})$ grid, where $t$ is time, $p$ is pressure, $z$ is altitude, $lon$ is longitude, and $lat$ is latitude. This is done using the retrieved ACE-FTS pressures, which are on a 1-km grid from 0.5 to 149.5 km. Since OSIRIS does not retrieve atmospheric pressure, the OSIRIS $O_3$, time, latitude, and longitude profiles (in altitude) are spline-interpolated to the ACE-FTS grid and assumed to have the same pressure values as their coincident ACE-FTS profile. Due to using the ACE-FTS pressures, this study can be considered to be estimating the natural variability on common pressure levels, rather than on common altitude levels.

For each profile, the model data are then linearly interpolated from the $(t_{MOD}, z_{INST}, lon_{MOD}, lat_{MOD})$ grid to a $(z_{INST}, lon_{MOD}, lat_{MOD})$ grid at $t_{INST}$. At each altitude, the $(lon_{MOD}, lat_{MOD})$ gridded data are then bilinearly interpolated to the $lon_{INST}$ and $lat_{INST}$ values at that altitude, using altitude dependent geolocations (e.g. Kolonjari et al., 2018). This leads to model $O_3$ data sampled at the instrument times and geolocations on a $(z_{INST}, t_{INST})$ grid. Outliers in the ACE-FTS data are filtered out using their quality flags, as per Sheese et al. (2015), and the corresponding data points are also removed from the corresponding OSIRIS and model data sets. The OSIRIS data were not filtered for outliers.

The estimated geophysical variability, as per the model data sets, was defined to be the $2\sigma$ standard deviation of the differences between simulated ACE-FTS values and simulated OSIRIS values (at each altitude),

$$v_{geo} = 2 \times stdev(MOD^{ACE} - MOD^{OS}). \tag{1}$$

In relative terms, the relative differences are calculated as the differences between ACE-FTS and OSIRIS divided by the overall mean of all ACE-FTS and OSIRIS values at that altitude,

$$rel\ diff_i = 2N \frac{MOD_i^{ACE} - MOD_i^{OS}}{\sum_j^N MOD_j^{ACE} + MOD_j^{OS}} \times 100\%, \tag{2}$$

where $N$ is the number of coincident values at that altitude. The overall mean in the denominator was used in order to be consistent with Sheese et al. (2016; 2017), where it was used to minimize the effect of retrieved negative values. The relative geophysical variability was calculated as the $2\sigma$ standard deviation of the relative differences. The same equations were used for determining the relative differences and the $2\sigma$ variations between the actual ACE-FTS and OSIRIS measurements (replacing MOD in Eqs. 1 and 2 with INST).

## 4 Results

### 4.1 Global comparisons

Coincidence criteria of within 6 h and 500 km were first chosen, yielding the profiles of mean $O_3$ bias (ACE-FTS – OSIRIS) due to sampling and geophysical variability profiles ($2\sigma$ variation) shown in Fig. 1. Also shown are the profiles of the actual measurement bias and $2\sigma$ variation of the differences at those criteria. All three models exhibit a small bias (within 20 ppbv, 0.5%) between 12 and 29 km. Between 30 and 45 km, the model results indicate that ACE-FTS $O_3$ values are expected to be systematically lower than OSIRIS. CMAM indicates a bias of up to 23 ppbv (0.5%) in this region, EMAC indicates a bias of up to 64 ppbv (1.1%), and up to 0.13 ppmv (2.8%) with WACCM. Above 48 km, all three models exhibit systematically larger concentrations of ACE-FTS $O_3$ than OSIRIS $O_3$. EMAC indicates a bias of up to 25 ppbv (2.1%) in this region, CMAM indicates a bias of up to 49 ppbv (3.9%), and up to 0.10 ppmv (8.7%) with WACCM. The more extreme values yielded by the WACCM simulations could in part be due to the finer horizontal resolution.

All three models agree well in terms of geophysical variability. In absolute terms, all three profiles of $2\sigma$ variation increase from ~0.1 ppmv at 10 km, to on the order of 0.5-0.6 ppmv near 30-40 km, then decrease with altitude to ~0.2 ppmv near 55 km. In relative terms, all three decrease from within 27-32% near 10 km to 7-9% near 21 km. Between 21 and 52 km, the

simulated geophysical variability profiles are typically on the order of 7-11%, with WACCM exhibiting the largest variability of 12% at 42 km. Above 52 km, variability increases with altitude to 10-12% at 55 km.

In order to estimate the uncertainty introduced by model sampling uncertainties (interpolation uncertainties, and uncertainties introduced by assuming ACE-FTS altitude-pressure values for OSIRIS), the standard run WACCM data that were linearly interpolated in time and bilinearly-interpolated to the measurement geolocations were compared with WACCMOL profiles (i.e., profiles from a WACCM run with output directly at the satellite observation times and geolocations). In this specific case, both WACCM and WACCMOL assumed altitude-independent geolocations (30-km tangent height values). Figure 2 shows the $2\sigma$ variability between coincident ACE-FTS and OSIRIS $O_3$ profiles as determined by WACCM and WACCMOL at coincidence criteria of within 6 h and 500 km. The difference in geophysical variability between WACCM and WACCMOL is typically within $\pm1\%$ between 11 and 38 km and within $\pm2\%$ between 10 and 47 km. Above 47 km, the difference increases sharply up to 7% near 55 km, however between 30 and 55 km that difference in absolute terms is on the order of 0.04-0.06 ppmv. These results suggest that in the upper stratosphere the interpolation method may be underestimating the magnitude of the geophysical variation.

Simulated geophysical variability can also be determined for a range of coincidence criteria. Figure 3 shows the geophysical variability determined from CMAM, EMAC, and WACCM for all time difference criteria between within 1.5 h and within 12 h in 0.5 h increments and distance difference criteria between within 150 km and within 2000 km in 50 km increments. These were calculated for all three models at all altitude levels (10-55 km), and results are shown for altitude levels of 20.5, 40.5, and 55.5 km.

Again, all three models show very similar geophysical variability patterns for different coincidence criteria. At the lowest altitudes (e.g., 20.5 km), where there are relatively small diurnal variations, for any given distance criterion, geophysical variability tends to stay fairly constant regardless of the time criterion (up to within 12 h). Conversely, for any given time criteria, geophysical variability increases from ~2-7% at within 150 km to ~16-23% at within 2000 km. At the highest altitudes (e.g., 55.5 km), the opposite effect is seen. Since there is a significant diurnal effect, the simulated geophysical variability is fairly consistent at a given time criterion, regardless of the distance criterion; and at any given distance criterion, the geophysical variability typically increases from ~6-12% at within 1 h to 13-22% at within 12 h. At intermediate altitudes (e.g. 40.5 km), where there is a moderate diurnal cycle, the geophysical variability tends to increase with both time and distance criteria. The variability increases from ~2-5% near within 1 h and 100 km to ~12-15% near within 12 h and 2000 km.

The mean of all three model results was taken to give ensemble mean values of the geophysical variability, shown in Fig. 4. These closely resemble the results described above, with geophysical variability being relatively independent of the time difference criterion at the lower altitude levels, relatively independent of the distance difference criterion at the higher altitude levels, and dependent on both at the intermediate altitude levels. When comparing $O_3$ measurements between ACE-FTS and OSIRIS, these ensemble mean geophysical variability values can be used to optimize coincidence criteria. At each altitude level, "optimized" coincidence criteria can be chosen where there are the greatest number of coincident measured profiles with the estimated geophysical variability less than a desired value. For instance, the circle markers on the plots in Fig. 4 indicate

"optimized criteria" where there are the greatest number of coincident ACE-FTS and OSIRIS profiles when the estimated geophysical variability is less than 10%, and Fig. 5 shows results for comparisons between ACE-FTS and OSIRIS $O_3$ profiles when using the "optimized criteria" for this chosen 10% $2\sigma$ variability limit at each altitude. It should be noted that in Fig. 5, at some of the altitude levels below 17 km there were no coincidence criteria evaluated where geophysical variability was less than 10%, and in those cases coincidence criteria of within 1.5 h and 150 km were used. The coincidence criteria can be optimized for any chosen limit of geophysical variability (10% was chosen in this case), and naturally this could be done for any subset of seasons/latitudes within the collocated data. Although, one drawback to having different coincidence criteria at each altitude, especially when making global comparisons, is that it can potentially add biases between altitudes due to changing seasonal and latitudinal sampling. Therefore, care must be taken to ensure that biases of this type are not being introduced.

These results can be used not only to constrain the inherent geophysical variability in comparisons between satellite measurements but also increase the number of usable coincident profiles. Figure 6 shows results of comparisons between ACE-FTS and OSIRIS $O_3$ profiles for five different coincidence criteria: within 2 h and 250 km, within 6 h and 500 km, within 8 h and 1000 km, within 12 h and 2000 km, and criteria optimized at each altitude. The optimized criteria were such that above 17 km the maximum estimated geophysical variability was 10% and below 17 km it was 15%. At most altitudes, the bias between the two instruments is relatively independent of coincidence criteria and the profiles exhibit similar variations with altitude. Above 20 km, the differences between the biases given different coincidence criteria are typically on the order of 1-4%. These differences are slightly larger below 20 km, where the maximum difference is 8% between the 2 h and 250 km criteria and the 12 h and 2000 km criteria. The $2\sigma$ standard deviations of the relative differences, shown in Fig. 6b, exhibit greater variability with coincidence criteria. Between 20 and 40 km, the optimized criteria yield standard deviations that are typically better than all the other criteria, with the exception of within 2 h and 250 km below 14 km and between 20 and 42 km. However, with the criteria of 2h and 250 km only 279 coincident profiles (Fig. 5c) are being compared, whereas with the optimized criteria, 1900-5900 profiles are used in the comparisons, leading to a more robust result and with a consistent estimate on the geophysical variability uncertainty. The increase in coincident profiles may not be necessary in this exact case where global data are being compared, but would be useful in specific regions where there are fewer coincident profiles with which to compare. The greatest improvement to the standard deviations is in the 13-20 km region, where the optimized criteria lead to standard deviations on the same order as the 2 h and 250 km criteria, but making use of 2-7 times more profiles, and, again, providing an estimate on the geophysical variability uncertainty.

## 4.2 Hemispheric comparisons

It is also interesting to observe the difference in geophysical variability between the polar NH (poleward of 50°N) region and the polar SH (poleward of 50°S) region, where there is greater $O_3$ variability in general. Fig. 7 shows the same plots as those of Fig. 4, but for polar data binned by hemisphere and month, and it shows that, unsurprisingly, there is a much larger difference between polar NH and SH is in the stratosphere during the end of winter than at the beginning of summer. This is due to the

stronger Southern polar vortex. At 18 km, at coincidence criteria of within 8 h and 1000 km, the ensemble mean geophysical variability in the NH is 15%, whereas in the SH $O_3$ concentrations are estimated to be over twice as variable, at 35%. Figure 8 shows the difference in ensemble mean geophysical variability at coincidence criteria of 8 h, 1000 km.

Above 15 km in the summer months, when there is not a strong polar vortex, the NH and SH exhibit similar geophysical variability profiles, with variability on the order of 5-15%. In the same altitude region in the SH spring, geophysical variability is much larger, due to the strong and prevalent Southern polar vortex, which is just starting to break up with the onset of sunlight; and at laxer coincidence criteria, it is more likely that one instrument will be observing inside the Southern polar vortex and the other outside the vortex, which can have different atmospheric conditions. The variability is on the order of 15-20% above 22 km, and peaks at 35% near 18 km, where there is some of the most ozone depletion. As can be seen on the left panel of Fig. 7, in the lower stratosphere in the polar SH, the geophysical variability is more sensitive to the time coincidence criterion than in the polar NH. The NH geophysical variability above 30 km is also greater at the end of winter (~5-10%) than during the summer (~10-15%). This could be due to stronger planetary wave forcing in the NH (e.g. Butchart, 2014; de la Cámara, 2018) and/or stronger descent of NO and $NO_2$ following sudden stratospheric warming events (e.g. Reddmann et al., 2010).

Figure 9 shows the estimated bias and the mean $2\sigma$ variability of the relative differences between ACE-FTS and OSIRIS $O_3$ profiles in the polar SH for different coincidence criteria, including the optimized criteria for 10% geophysical variability above 17 km and 15% below 17 km. As with the global comparisons, the bias is largely unaffected by the choice of coincidence criteria. Between 20 and 42 km, all the coincidence criteria lead to similar variability profiles on the order of 15-20%. Below 20 km, the optimized criteria tend to yield better variability results than the criteria of within 6 h and greater; although, they yield larger values than the 2 h and 250 km criteria. The benefits of the optimized criteria case in this region are that there is a consistent estimate of the geophysical variability and that it makes use of more coincident profiles—the 2 h and 250 km criteria have a maximum of only 54 profiles, whereas the optimized criteria make use of up to 307 profiles near 15 km.

## 5 Summary

This study used three different chemistry-climate models—CMAM, EMAC, and WACCM—that were run in specified dynamics mode, i.e. meteorological fields were nudged towards observational data. The $O_3$ data from these models were sampled at ACE-FTS and OSIRIS times and locations in order to estimate the geophysical variation (as characterized by the $2\sigma$ standard deviation of differences) inherent in the satellite $O_3$ comparisons at varying coincidence criteria. The averages of the simulated values were taken in order to obtain ensemble mean values of the geophysical variation. Based on the differences in the estimated geophysical variation between WACCM and WACCMOL (WACCM output at Observed Locations), the interpolation method used in this study yields the most accurate results in the lower to mid stratosphere, up to ~25 km. Above 30 km the interpolation may lead to an underestimation of the geophysical variability on the order of 0.04-0.06 ppmv (a relative difference of up to 23%).

When analysing the global data, all three models show similar geophysical variability patterns based on coincidence criteria. In the lower stratosphere, the geophysical variation is, within the criteria limits, relatively independent of the time criterion and increases as the distance criterion is widened. In the upper stratosphere, where there is a stronger $O_3$ diurnal cycle, the geophysical variation tends to be independent of the distance criterion and increases when the time criterion is increased. In the middle stratosphere, the geophysical variation tends to increase with increasing time and distance criteria. Ensemble mean values in the lower stratosphere show that geophysical variability is much larger in the high-latitude SH than in the high-latitude NH, except at very tight criteria (e.g. within 2 h and 200 km). This is due to the more consistent presence of the Southern polar vortex, which often leads to coincident ACE-FTS and OSIRIS measurements sampling two different air masses (inside and outside the vortex). In the NH, geophysical variation decreases more strongly with altitude from 24% at 12 km to 8% at 20 km, where as in the SH geophysical variation is 28% at 17 km and 20% at 20 km. Also, in the polar SH in the lower stratosphere, geophysical variability does not tend to be time independent.

When comparing profiles from satellite data, the ensemble means of the simulated geophysical variability can be used to optimize the chosen coincidence criteria, allowing for a large number of coincident profiles while limiting the estimated variability to a desired quantity (on the scale of the measurement uncertainties). This method allows for relatively simple, consistent estimates of geophysical variability inherent in the comparison results and allows for making use of more coincident profiles, which is an advantage for solar occultation instruments that tend to have fewer observation profiles than sensors using other limb viewing techniques. However, this does lead to different measurement times/locations being compared at different altitude levels, and therefore care must be taken such that it does not lead to regional and/or seasonal sampling differences in the profiles of the comparisons results, which could add spurious features. This technique of using the natural variability estimates in order to optimize the coincidence criteria can however also be used for data that is isolated to a single season/latitude range.

**Data availability**

The sampled data sets and simulations used for these analyses are available (doi:10.5683/SP2/ZHGQOI). The ACE-FTS Level 2 data can be obtained via the ACE-FTS website (registration required), http://www.ace.uwaterloo.ca. The OSIRIS data can be obtained via http://odin-osiris.usask.ca (registration required, doi:10.5281/zenodo.4110053). The CMAM30 data set can be downloaded via Environment and Climate Change Canada's climate modelling website, http://climate-modelling.canada.ca/climatemodeldata/cmam/output/.

**Author contribution**

Author contributions: The study was designed by PES, TvC, and KW. PES wrote the manuscript. PES performed the analyses with contributions from FK. Satellite data used in this study were provided by CDB, and DAD. Model simulations used in the

study were provided by DP, DEK, and PJ. Valuable comments on the manuscript were provided by KW, CDB, FK, DAD, DP, DEK, PJ, and TvC.

**Competing interests**

TvC is Associate editor of AMT but he has not been involved in the evaluation of this manuscript.

5 **Acknowledgements**

This project was funded by the Canadian Space Agency (CSA). The Atmospheric Chemistry Experiment is a Canadian-led mission mainly supported by the CSA. We thank Peter Bernath for his leadership of the ACE mission. Odin is a Swedish-led satellite project funded jointly by Sweden (Swedish National Space Board), Canada (CSA), France (Centre National d'Études Spatiales), and Finland (Tekes), with support by the third party mission programme of the European Space Agency (ESA). 10 The ACE-FTS height-dependent latitudes and longitudes were obtained from the geolocation files on the ACE-FTS website (database.scisat.ca; registration required); the ACE-FTS level 2 $O_3$ data were also obtained from that site. The OSIRIS level 1 height-dependent times, latitudes, and longitudes were obtained from the OSIRIS website (odin-osiris.usask.ca), as were the level 2 $O_3$ data. We thank the CSA for financial support that made the development of the CMAM30 dataset possible. The CMAM30 dataset can be downloaded from climate-modelling.canada.ca/climatemodeldata/cmam/cmam30/index.shtml. The 15 EMAC simulations have been performed at the German Climate Computing Centre (DKRZ) through support from the Bundesministerium für Bildung und Forschung (BMBF). DKRZ and its scientific steering committee are gratefully acknowledged for providing the HPC and data archiving resources for this consortial project ESCiMo (Earth System Chemistry integrated Modelling). WACCM is a component of the Community Earth System Model (CESM), which is supported by the National Science Foundation. We would like to acknowledge high-performance computing support from Cheyenne 20 (doi:10.5065/D6RX99HX) provided by NCAR's Computational and Information Systems Laboratory, sponsored by the National Science Foundation. We thank NASA Goddard Space Flight Center for the MERRA data (available freely online at disc.sci.gsfc.nasa.gov).

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

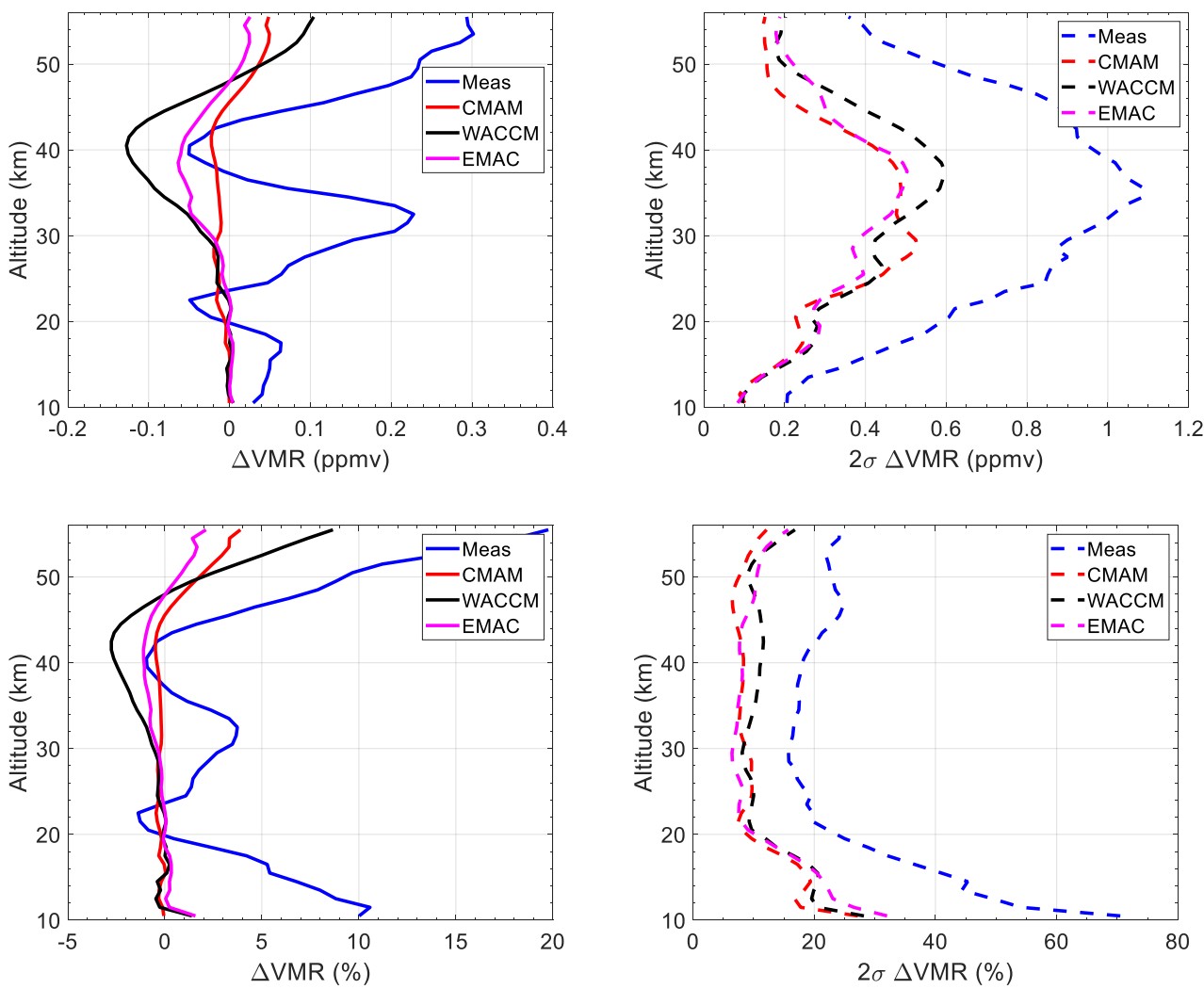

**Figure 1. Measured and simulated mean differences (left) between ACE-FTS and OSIRIS O$_3$ and the corresponding 2σ variability (right). Using profiles for all available times and latitudes with coincidence criteria of within 6 h and 500 km. Results shown for the differences (top) and relative differences (bottom).**

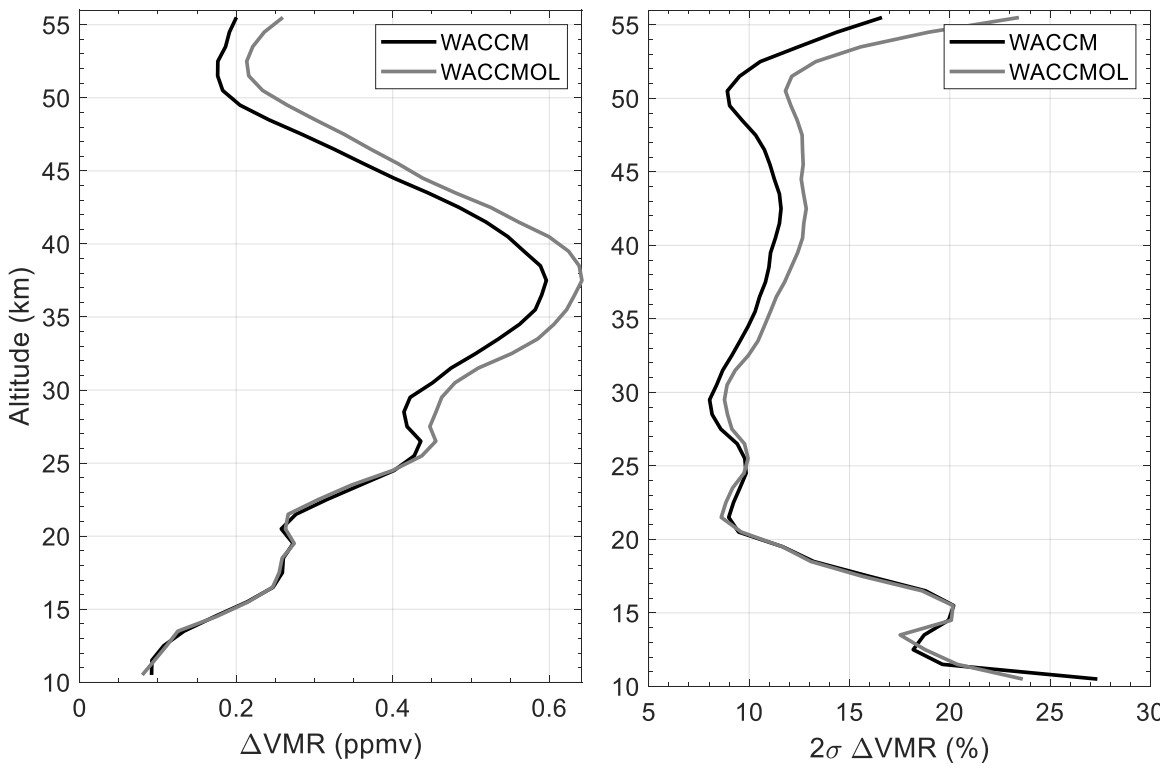

**Figure 2. Simulated 2σ variability (left) and relative 2σ variability (right) for ACE-FTS – OSIRIS coincident O$_3$ profiles when interpolating to measurement geolocations from WACCM grid (black) and using WACCMOL (WACCM output at Observed Locations; grey). Coincidence criteria of within 6 h and 500 km.**

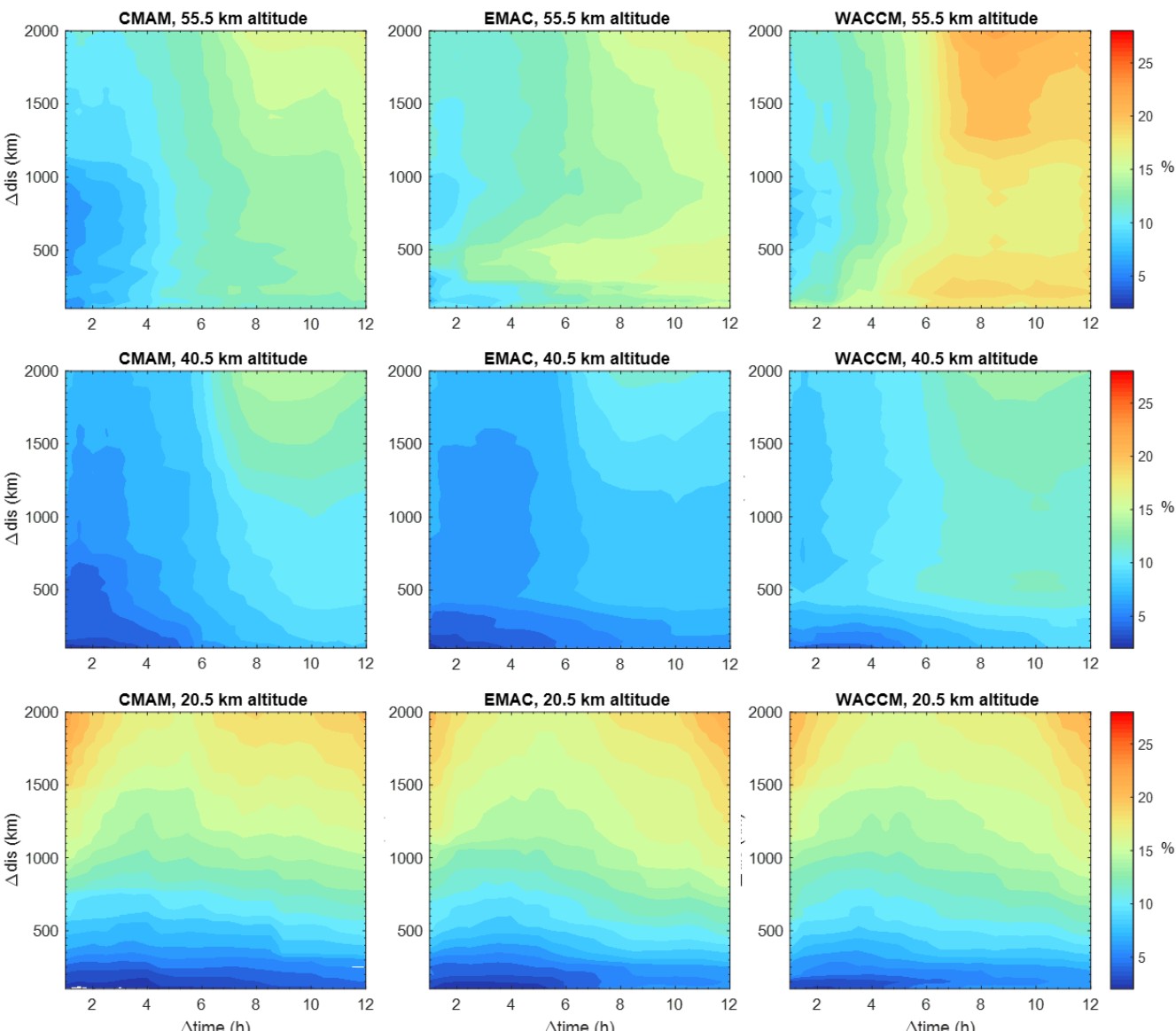

**Figure 3. Geophysical variability (2σ) between ACE-FTS and OSIRIS O₃ derived from the simulated results of CMAM (left), EMAC (centre), and WACCM (right), at altitudes of 20.5 km (bottom), 40.5 km (middle), and 55.5 km (top). Calculations performed for time difference criteria of within 1.5 h to within 12 h in 0.5 h increments and distance difference criteria of within 150 km to within 2000 km in 50 km increments.**

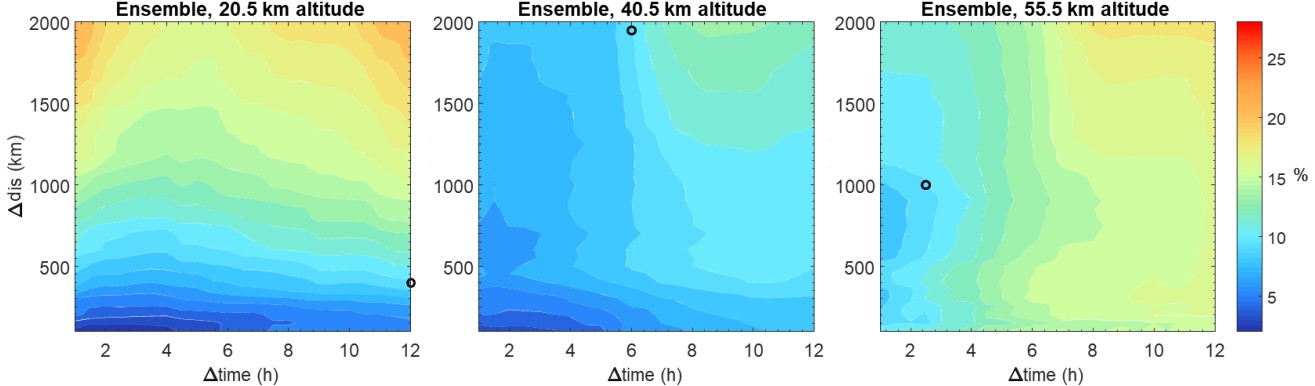

**Figure 4. Ensemble mean geophysical variability (2σ) between ACE-FTS and OSIRIS O₃, as estimated from CMAM, EMAC, and WACCM data. Calculations performed for time difference criteria of within 1.5 h to 12 h in 0.5 h increments and distance difference criteria of within 150 km to 2000 km in 50 km increments. Black circles indicate the coincidence criteria optimized for the greatest number of coincident profiles with geophysical variability limited to 10%.**

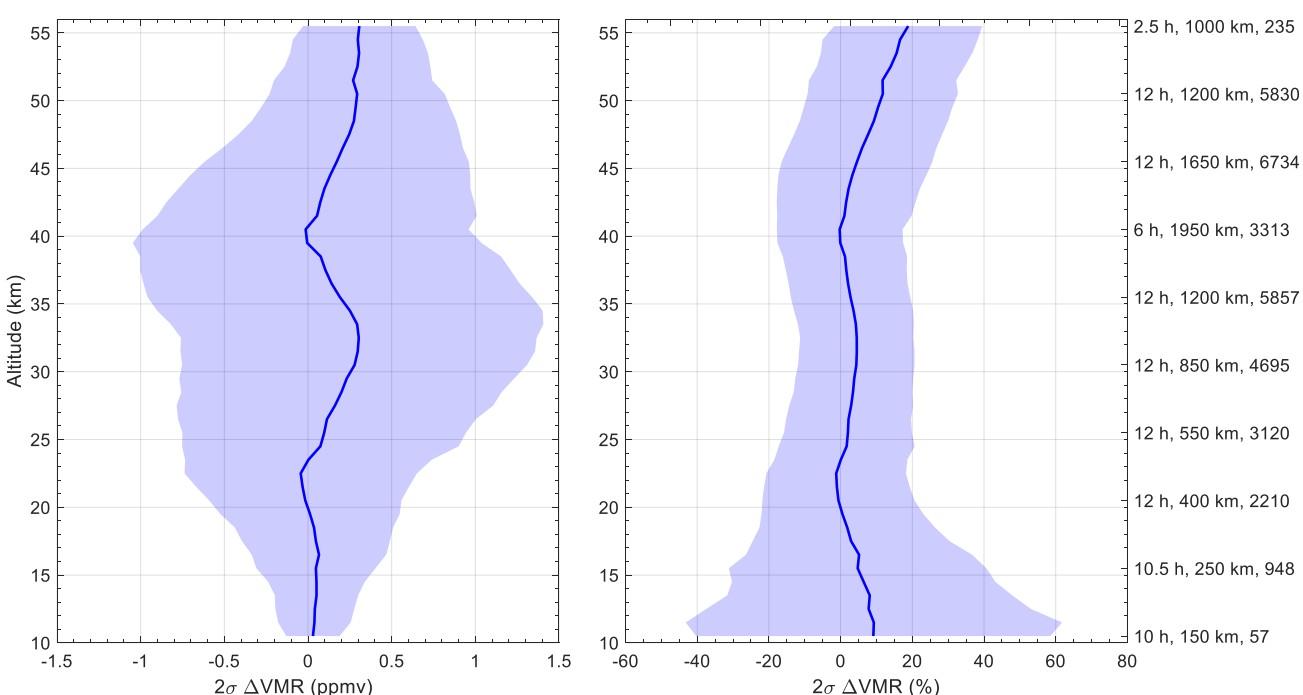

**Figure 5. Comparisons between ACE-FTS and OSIRIS O₃ profile measurements at coincidence criteria that, at each altitude, maximize the number of coincident profiles while keeping estimated geophysical variability below 10%. Solid lines indicate the mean of the differences (left panel: absolute values; right panel: relative differences), and shaded regions are the corresponding 2σ variations from the means.**

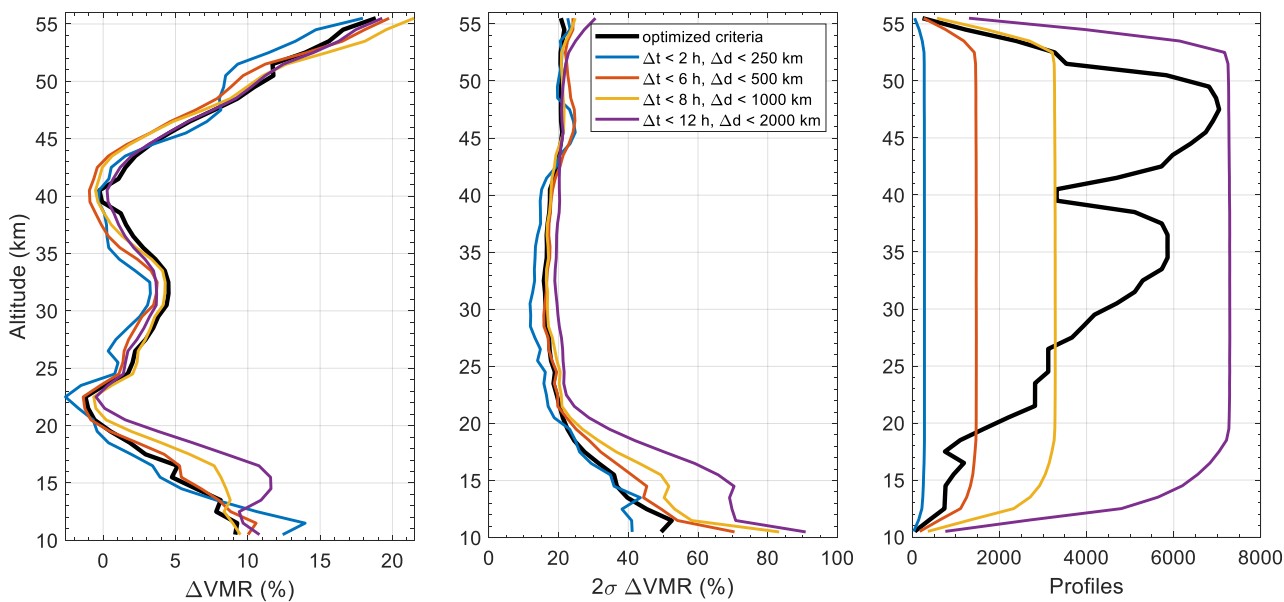

**Figure 6. Comparison results between ACE-FTS and OSIRIS O₃ for different coincidence criteria, (left) mean of the relative differences, (centre) 2σ variation of the relative differences, and (right) number of coincident profiles. Optimized criteria are for less than 10% geophysical variability above 17 km and less than 15% below 17 km.**

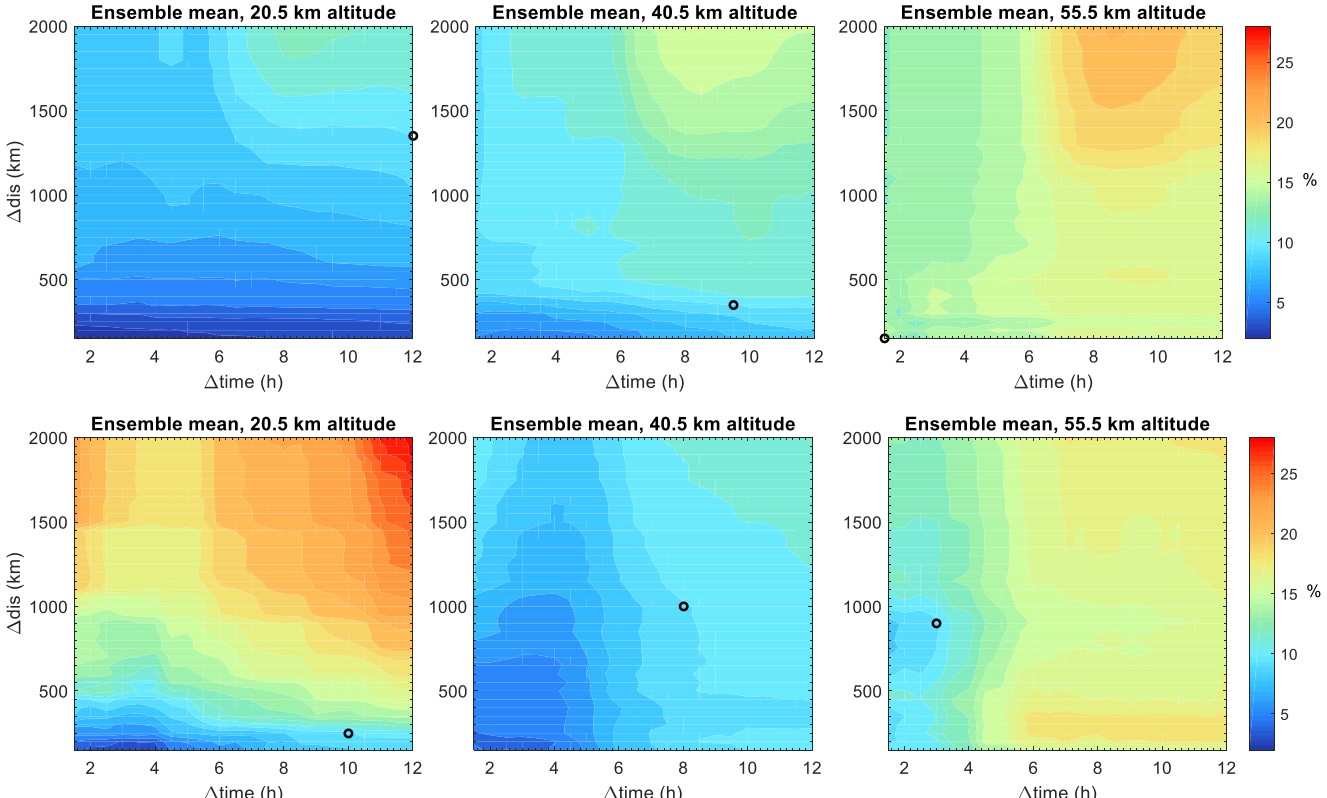

**Figure 7. Ensemble mean geophysical variability (2σ) between ACE-FTS and OSIRIS O₃, as estimated from CMAM, EMAC, and WACCM data, (top) for 50-90°N, and (bottom) for 50-90°S. Calculations performed for time difference criteria of within 1.5 h to within 12 h in 0.5 h increments and distance difference criteria of within 150 km to within 2000 km in 50 km increments. Black circles indicate the coincidence criteria optimized for the greatest number of coincident profiles with geophysical variability limited to 10%.**

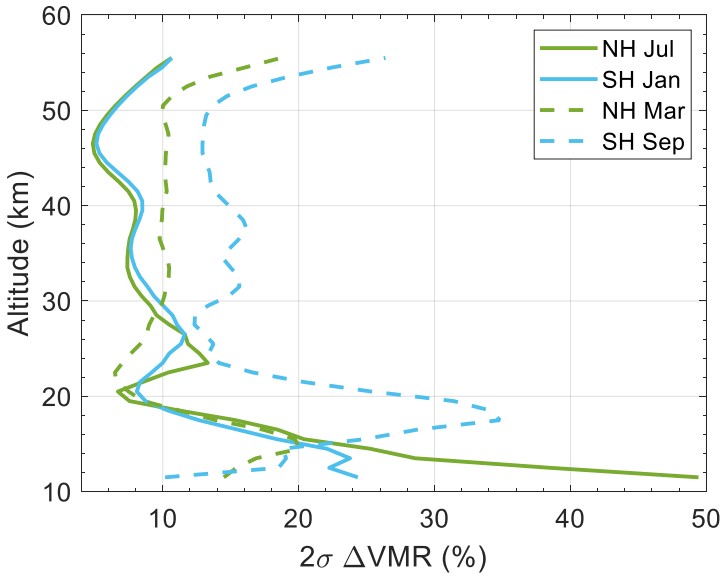

**Figure 8. Ensemble mean geophysical variability (2σ) between ACE-FTS and OSIRIS O$_3$ in the polar regions at coincidence criteria of within 8 h and 1000 km, as estimated from CMAM, EMAC, and WACCM data.**

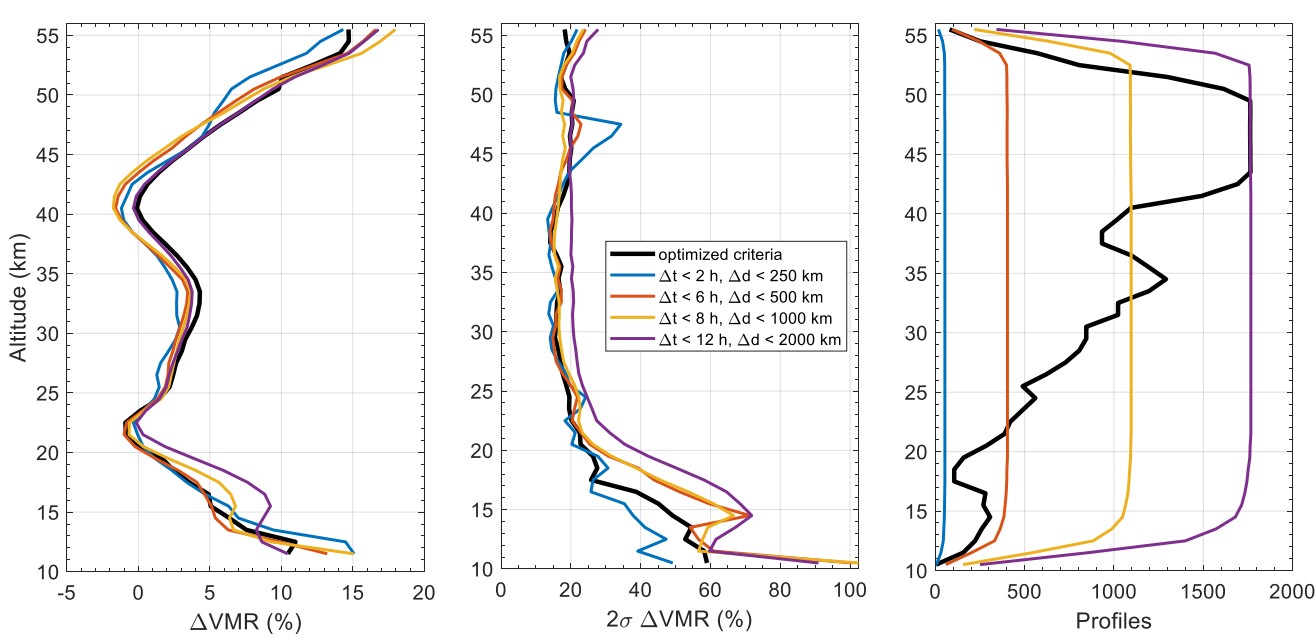

5    **Figure 9. The 2σ variability of the relative differences between ACE-FTS and OSIRIS O$_3$ profiles in the Southern polar region at different coincidence criteria, including the optimized criteria for 10% variability above 17 km and 15% below 17 km (left), and the corresponding number of coincident profiles (right).**