# Peer review of "Model estimations of geophysical variability between satellite measurements of ozone profiles"

_Atmospheric Measurement Techniques, 2020_

## Referee Comment (RC1) · Anonymous Referee #1 · 5 Aug 2020

This manuscript investigates the geophysical variability encounter in the comparison of ACE-FTS and OSIRS for different coincident criteria (the spatiotemporal window used to define a coincidence). Geophysical variability is estimated by sampling at the measurement's times and locations 3 model fields (WACCM, CMAM, and EMAC). There are major points that need to be addressed before publication.

Comments:

Page 2 line 24: The authors could include: Toohey et al 10.1002/jgrd.50874 and Millan et al 10.5194/acp-16-11521-2016. Both of these papers characterized sampling biases for ACE-FTS

[Figure]

Page 4 line 32: This section "Another set of output from this WACCM simulation was used in this study, with the same setup, the only difference being that the output model data were sampled at the time and geolocations (with WACCM altitude profiles) of the ACE-FTS and OSIRIS observation profiles (individual profiles were assumed to be at a single time, latitude, and longitude, taken as the 30-km tangent height values). The WACCM data output at the instrument Observed Locations will from here onward be referred to as WACCMOL." is a bit confusing. The way is written, the reader may wrongly infer that, WACCMOL refers to WACCM at (tinst, Zmod, loninst, latinst) since that is what the authors were describing right before. Please clarify this section.

Section 2.3, Please give examples / citations of how good or bad these models capture the geophysical variability. Or give examples of the type of phenomena that have been study with these models.

Page 5 line 13: If I understand this phrase correctly "First, for every instrument profile, the model O3 data closest in time to tinst on either side are isolated and are spline-interpolated in log-space from the native"

It implies that only one side (i.e. only one synoptic time) is used during the inter-polations. If that's the case, the interpolation in time is actually extrapolating. The interpolation should be performed as follows:

(tmod, Pmod, lonmod and latmond) grid to a (tmod, Zinst, lonmod and latmond) at the two closest times encompassing the measurement, that is at both sides of the measurement time.

Then interpolate to (tinst, Zinst, lonmod and latmond) using (t0, Zinst, lonmod and latmond) and (t1, Zinst, lonmod and latmond)

and lastly to (tinst, zinst, loninst, latinst)

If that is not the case, i.e. I misinterpret the phrase, please clarify the text, perhaps something like: "First, for every instrument profile, the model O3 data closest in time to

tinst on both sides are isolated and are spline-interpolated in log-space from the native . . ."

Page 5 line 16, with respect to: "Since OSIRIS does not retrieve atmospheric pressure, the OSIRIS O3, time, latitude, and longitude profiles (in altitude) are spline-interpolated to the ACE-FTS grid and assumed to have the same pressure values as their coincident ACE-FTS profile." First off, in the model to model comparisons there should not be any usage of the ACE-FTS pressure everything should be done using the model pressure (if needed at all). Secondly, in the INST comparisons, the authors should not use the ACE-FTS pressure for OSIRIS, they should use the interpolated pressure and temperature profiles obtained from the European Centre for Medium-Range Weather Forecasts during the OSIRIS retrieval process. If those pressure levels are not available, at least interpolate the CMAM30SD pressure (which in essence is ERA-Interim) to the OSIRIS measurement times and locations; that would be a much realistic comparison.

Page 5 line 30 (equation 2): Please clarify why are the authors using the overall mean of all ACE-FTS and OSIRIS values at a given altitude in the denominator as opposed to just (MODace_i + MODos_i). That is, why not simply use

reldiff_i = 2N (MODace_i – MODos_i) / (MODace_i – MODos_i) * 100.

which is what is most commonly used for example in validation papers (for example in Dupuy et al 2009 doi:10.5194/acp-9-287-2009 or Bognar et al 2019 doi:10.1016/j.jqsrt.2019.07.014 ACE-FTS validation papers.

Page 6 line 6 (fig 1): Not all models yield profiles with similar patterns in the mean o3 bias; CMAM and EMAC maybe but WACCM shows a clear departure, while CMAM and EMAC mean biases are well within ~1% from 10 to 48km, WACCM can be as low as -3%. This may not sound much but its departure from the other 2 models is clearly visible, is this because the variability in WACCM is greater than that found in the other 2 models.

Also, now that the geophysical variability has been estimated, could you add it to the expected instrument noise variability and see if you can get back the measured variation (the blue line). This assumes that the covariance between ozone and instrument noise is zero, which it presumably is.

Further, how come the bias found in this figure is not similar to the one found in Dupuy et al 2009, figure 7 or the one found in Bognar et al 2019 Figure 2. Both comparing ACE-FTS and OSIRIS. Please mention how the ACE-FTS-OSIRIS biases found on this study compare with the others validation papers and explain any differences.

In page 6 line 20: If WACCM is (tinst, Zmod, loninst, latinst) and WACCM-OL is (tinzt, Zinst, loninst, latinst) there should not be any differences between the coincident ACE-FTS and OSIRIS O3 profiles as determined by WACCM and WACCMOL. Because WACCMOL should be able to be computed directly from the WACCM coincidences, since in the end, you should only need to interpolate from the geophysical variation at Zmod to the geophysical variation at Zinst, which should not induce any bias.

Further, during this test there should not be any usage of the ACE-FTS pressure. The authors are just comparing model values, interpolated to different times and locations (and altitudes for the OL). Hence, the bias points to a bug in the interpolation scheme. and the phrase "and uncertainties introduced by assuming ACE-FTS altitude-pressure values for OSIRIS" should be deleted. If the authors are using pressure at any point of this comparison, they should use the WACCM pressure interpolated either to the times and locations or to the times, locations and altitudes.

Page 7 line 4 (figure 3): It would be interesting to see the results for a 10 or 12 km, where a lot of geophysical variability will be found due to the location of the tropopause.

In page 7 line 22 (or more easily in Figure 5), the optimized criteria is chosen for geophysical variability is less than 10%. However according to page 2 line 6 "Collocated measurements should be close to each other relative to the spatiotemporal scale on which the variability of the geophysical field becomes comparable to the measure-

ment uncertainties", shouldn't then the optimized criteria be for less than the combined measurement uncertainties. According to section 2.1, "on the order of a few percent for ACE-FTS" (please be more specific) and according to section 2.2 within 5% for OSIRIS. Is the combined measurement uncertainty less than 10%, is using the combined measurement uncertainty as criteria to strict? so that only a few coincidences are found? What are the implications of this?

Also, please add in figure 5 the difference in percent so that it is easily comparable with the rest of the figures.

In page 7 line 24-26, the manuscript will be enriched showing an example of the biases that can be induced by having different coincidence criteria per height. Please quantify it.

Page 8 line 14: after "It is also interesting to observe the difference in geophysical variability between the polar NH (poleward of 50°N) region and the polar SH (poleward of 50°S) region" please add: where most of the ozone variability can be found.

I suggest splitting the analysis of Figure 8 into two periods, during polar vortex season and during the offseason. That way, the sentences about stronger descent, SSWs, and in or out the vortex will be more certain. And it will presumably showcase that the coincidence criteria to maintain a given geophysical variability criteria will vary with season.

Also, consider exploring the tropics separately where the tropopause is higher implying difference criteria for lower (tropospheric) altitudes.

Please add the mean of the relative differences panel to figure 9. So that Figure 6 and Figure 9 have the same layout.

Summary or all the manuscript really: Please make clear that the optimize criteria discussed are only valid for the ACE-FTS and OSIRIS pairings used in this study. That is, that a similar analysis will have to be conducted for all pair of instruments to be

compared.

---

## Referee Comment (RC2) · Anonymous Referee #2 · 9 Sep 2020

The paper discusses the important topic, influence of geophysical variability in data validation. To estimate this influence, the data from three chemistry-transport models have been used. Ozone profile data from two satellite instruments, ACE-FTS and OSIRIS, are considered in the paper. The interesting analyses are presented in the paper. However, I have several concerns on the methods and analyses. My comments are below.

MAIN COMMENTS

1) The authors characterize the natural variability of the ozone field using the data from 3 CCM/CTM models having rather low spatial resolution, from 1.9 deg to 3.75deg.

[Figure]

This implies that the lowest spatial scale that can be probed with these models is ~200-300 km. The smaller-scale variability is not resolved by the models and cannot be characterized. This should be at least stated clearly in the paper (in particular, p.2 line 29, "large scale" should be quantified). However, the optimal way would be the inclusion of simulations with a high-resolution CTM.

2) The characterization of ozone variability assessed using the model data is too simplified – both in general and for the considered application of satellite data validation. First, the variability ozone variability depends on latitude and season (in addition to diurnal variability). Second, the variability is not isotropic in latitude-longitude direction therefore a simple characterization by "separation distance" is too superficial. Using the model data, the spatio-temporal variability of ozone field can be characterized in more detail, and thus the collocation criteria can be used in more advanced way (see also below).

3) The idea of collocation criterion using the information about the natural variability obtained from modelling is good. It should be described in more detail how technically the collocation criterion "variability <10%" is applied. Do I understand correctly that you use time-space collocation criteria as shown by circles in Figure 4, i.e., these are globally for each altitude level?

My main concern is that 10 % threshold is not actually optimized. Why have you selected 10% as a threshold? It seems to be significantly larger than uncertainties of each satellite dataset, thus the objective stated in the introduction, "Collocated measurements should be close to each other relative to the spatiotemporal scale on which the variability of the geophysical field becomes comparable to the measurement uncertainties" is not satisfied. On other hand, in the tropical middle stratosphere, for example, the overall variability is ~5%, thus the criterion <10% variability will be satisfied automatically for any collocation criteria.

I think that the maximization of number of collocations within a variability window is not

the best approach, since selecting a broader spatio-temporal window increase both number of collocations and natural variability. Instead, reduction (or minimization) of uncertainty of the bias estimates (which depends on measurements uncertainties, natural variability, and number of collocations) would be a more concrete objective, and the advantages of "optimized" criteria can be quantified (for example, reduction of bias uncertainty from x% to y%).

Since the ozone variability strongly depends on location/season, it is expected that the optimized collocation criteria will also depend on location and season, or, at least, characterized into "low" and "high" ozone variability. At the same time, this would be reduce the drawback that you mentioned in the paper on page 7: " One drawback to having different coincidence criteria at each altitude is that it can potentially add biases between altitudes due to changing seasonal and latitudinal sampling".

DETAILED COMMENTS

1) P.2 , l. 29: please quantify "large scale" term

2) Section 2. Please add estimates of random uncertainties of ACE-FTS and OSIRIS ozone profiles.

3) Section 2.2. Why don't you use version 5.10, which, as you explained in the paper, is better than v.5.07?

4) P.5, lines 17-19: You use rather relaxed collocation criteria (12 h and 2000 km); what is the difference in pressure profiles for large separations and how this affects transformation of OSIRIS data to pressure grid? Is the pressure-altitude conversion using reanalysis data at OSIRIS locations less accurate?

5) Why do you define sigma_nat as 2* std (MOD_ace-MOD_OSIRIS)? One would expect sigma_nat= std (MOD_ace-MOD_OSIRIS)/sqrt(2).

6) I suggest revision of Section 4, according to the MAIN COMMENTS 2 and 3. The spatio-temporal ozone variability (time, altitude, latitude, longitude, season) can be

in detail characterized using the model data. For optimization based on variability, I suggest a categorization at least of "low" and "high" ozone variability (alternatively, according to latitude zone and season). I suggest also quantitative estimates of validation improvement (for example, reduction of uncertainties of bias estimate, bias detectability, quality of the spread estimate) based on the optimized collocation criteria.

7) Section 4.2: The ozone variability in polar regions depends strongly on season. This should be taken into account in the analyses.
* * *

---

## Author Comment (AC1) · 14 Oct 2020

**Response to Reviewers - Model estimations of geophysical variability between satellite measurements of ozone profiles" by Patrick E. Sheese et al.**

*We'd like to thank the reviewers for their helpful comments. Here we address the main and specific comments of each reviewer, with their comments in black and our responses in green.*

**Referee #1**

Page 2 line 24: The authors could include: Toohey et al 10.1002/jgrd.50874 and Millan et al 10.5194/acp-16-11521-2016. Both of these papers characterized sampling biases for ACE-FTS

*These references have been added.*

Page 4 line 32: This section "Another set of output from this WACCM simulation was used in this study, with the same setup, the only difference being that the output model data were sampled at the time and geolocations (with WACCM altitude profiles) of the ACE-FTS and OSIRIS observation profiles (individual profiles were assumed to be at a single time, latitude, and longitude, taken as the 30-km tangent height values). The WACCM data output at the instrument Observed Locations will from here onward be referred to as WACCMOL." is a bit confusing. The way is written, the reader may wrongly infer that, WACCMOL refers to WACCM at (tinst, Zmod, loninst, latinst) since that is what the authors were describing right before. Please clarify this section.

*This has been further clarified in the text, now reading: "Another set of WACCM simulations was used in this study, with the same setup, the only difference being that the output model data were directly output at the ACE-FTS and OSIRIS observation times and geolocations (individual observation profiles were assumed to be at a single time, latitude, and longitude, taken as the 30-km tangent height values). The WACCM output at the instrument Observed Locations will from here onward be referred to as WACCMOL."*

Section 2.3, Please give examples / citations of how good or bad these models capture the geophysical variability. Or give examples of the type of phenomena that have been study with these models.

*Three or more references per model have been added. We also now state "All three models used in this study are considered to be 'state-of-the-art' stratosphere-resolving chemistry climate models and regularly participate in multi-model intercomparisons, including the exhaustive model assessments performed for CCMVal-2 (SPARC CCMVal, 2010) and CCMI-1 (Morgenstern et al., 2017)."*

Page 5 line 13: If I understand this phrase correctly "First, for every instrument profile, the model O3 data closest in time to tinst on either side are isolated and are splineinterpolated in log-space from the native"

It implies that only one side (i.e. only one synoptic time) is used during the interpolations. If that's the case, the interpolation in time is actually extrapolating.

That is not the case. It's interpolating between time steps (the two time steps that the instrument time is between). This has been made clearer in the text.

The interpolation should be performed as follows:

(tmod, Pmod, lonmod and latmond) grid to a (tmod, Zinst, lonmod and latmond) at the two closest times encompassing the measurement, that is at both sides of the measurement time.

Then interpolate to (tinst, Zinst, lonmod and latmond) using (t0, Zinst, lonmod and latmond) and (t1, Zinst, lonmod and latmond)

and lastly to (tinst, zinst, loninst, latinst)

If that is not the case, i.e. I misinterpret the phrase, please clarify the text, perhaps something like: "First, for every instrument profile, the model O3 data closest in time to tinst on both sides are isolated and are spline-interpolated in log-space from the native . . ."

*Sorry, the phrasing was confusing. The way you are suggesting is what was done. Your wording is now used in the text.*

Page 5 line 16, with respect to: "Since OSIRIS does not retrieve atmospheric pressure, the OSIRIS O3, time, latitude, and longitude profiles (in altitude) are spline-interpolated to the ACE-FTS grid and assumed to have the same pressure values as their coincident ACE-FTS profile." First off, in the model to model comparisons there should not be any usage of the ACE-FTS pressure everything should be done using the model pressure (if needed at all). Secondly, in the INST comparisons, the authors should not use the ACE-FTS pressure for OSIRIS, they should use the interpolated pressure and temperature profiles obtained from the European Centre for Medium-Range Weather Forecasts during the OSIRIS retrieval process. If those pressure levels are not available, at least interpolate the CMAM30SD pressure (which in essence is ERA-Interim) to the OSIRIS measurement times and locations; that would be a much realistic comparison.

*The model data are interpolated to ACE pressures, because the models are on pressure grids, not altitude grids. ACE was chosen because it provides both (retrieved) in the L2 files.*

*The ACE pressures were originally used out of convenience as OSIRIS pressures were not provided in the level 2 files. The interpretation can be considered as comparing what the models say the geophysical variation is between two common pressure levels as opposed to two common altitude levels. This is now stated in the text, "Due to using the ACE-FTS pressures, this study can be considered to be estimating the natural variability on common pressure levels, rather than on common altitude levels."*

*As a test, for a subsection of the data (CMAM for 2007-2008) we compared results of CMAM geophysical variability between the ACE and OSIRIS co-locations using only the ACE pressures and then the ACE and ECMWF pressures. The differences did not impact the conclusions..*

Page 5 line 30 (equation 2): Please clarify why are the authors using the overall mean of all ACE-FTS and OSIRIS values at a given altitude in the denominator as opposed to just (MODace_i + MODos_i). That is, why not simply use

reldiff_i = 2N (MODace_i – MODos_i) / (MODace_i – MODos_i) * 100.

which is what is most commonly used for example in validation papers (for example in Dupuy et al 2009 doi:10.5194/acp-9-287-2009 or Bognar et al 2019 doi:10.1016/j.jqsrt.2019.07.014 ACE-FTS validation papers.

*The past few ACE validation papers have used this method as a way to minimize the effects of not throwing out negative retrieved values. Negative values in the suggested methodology can cause percent differences to be unrealistically large. This has been mentioned in the text, "The overall mean in the denominator was used in order to be consistent with Sheese et al. (2016; 2017), where it was used to minimize the effect of retrieved negative values."*

Page 6 line 6 (fig 1): Not all models yield profiles with similar patterns in the mean o3 bias; CMAM and EMAC maybe but WACCM shows a clear departure, while CMAM and EMAC mean biases are well within ~1% from 10 to 48km, WACCM can be as low as -3%. This may not sound much but its departure from the other 2 models is clearly visible, is this because the variability in WACCM is greater than that found in the other 2 models.

*By "similar patterns" we didn't mean that all three profiles are exactly the same, which is why we discuss the difference later in this paragraph. We meant they are well correlated in altitude. "similar patterns" is now deleted.*

Also, now that the geophysical variability has been estimated, could you add it to the expected instrument noise variability and see if you can get back the measured variation (the blue line). This assumes that the covariance between ozone and instrument noise is zero, which it presumably is.

*Neither ACE-FTS nor OSIRIS have full uncertainty budgets that report the expected instrument noise on its own, and the focus of this paper is not the validation of the instruments, it's on geophysical variability.*

Further, how come the bias found in this figure is not similar to the one found in Dupuy et al 2009, figure 7 or the one found in Bognar et al 2019 Figure 2. Both comparing ACE-FTS and OSIRIS. Please mention how the ACE-FTS-OSIRIS biases found on this study compare with the others validation papers and explain any differences.

*The Dupuy paper used different versions of ACE-FTS and OSIRIS $O_3$ data. Bognar et al. (2019) focused on the high Arctic. The different bias profiles have the same shape, and the differences in magnitude can be explained by the fact that they are different versions, coincidence criteria and locations.*

In page 6 line 20: If WACCM is (tinst, Zmod, loninst, latinst) and WACCM-OL is (tinzt, Zinst, loninst, latinst) there should not be any differences between the coincident ACEFTS and OSIRIS O3 profiles as determined by WACCM and WACCMOL. Because WACCMOL should be able

to be computed directly from the WACCM coincidences, since in the end, you should only need to interpolate from the geophysical variation at Zmod to the geophysical variation at Zinst, which should not induce any bias.

*Both WACCM and WACCMOL are (tinst, Zinst, loninst, latinst). WACCMOL was a separate run with the output at the specific ACE-FTS times/geolocations. WACCM is the run on the standard grid, interpolated to the ACE-FTS times/geolocations. So, yes, this is a measure of how good the interpolation scheme is compared to a run that had the output at the exact location. This is now made clearer in the text, "In order to estimate the uncertainty introduced by model sampling uncertainties (interpolation uncertainties, and uncertainties introduced by assuming ACE-FTS altitude-pressure values for OSIRIS), the standard run WACCM data that were linearly interpolated in time and bilinearly-interpolated to the measurement geolocations were compared with WACCMOL profiles (i.e., profiles from a WACCM run with output directly at the satellite observation times and geolocations)."*

Further, during this test there should not be any usage of the ACE-FTS pressure. The authors are just comparing model values, interpolated to different times and locations (and altitudes for the OL). Hence, the bias points to a bug in the interpolation scheme. and the phrase "and uncertainties introduced by assuming ACE-FTS altitude-pressure values for OSIRIS" should be deleted. If the authors are using pressure at any point of this comparison, they should use the WACCM pressure interpolated either to the times and locations or to the times, locations and altitudes.

*Strictly speaking, you're correct, we wouldn't need to ACE pressures if we wanted to compare the output simply on the model pressure grid. But, we use the ACE pressures, because we want the output on the instrument altitude grid. "and uncertainties introduced by assuming ACE-FTS altitude-pressure values for OSIRIS" has not been deleted because that is part of what the figure is showing, since the WACCM data uses the ACE pressures for the OSIRIS locations because this is the grid we are using for the comparisons.*

Page 7 line 4 (figure 3): It would be interesting to see the results for a 10 or 12 km, where a lot of geophysical variability will be found due to the location of the tropopause.

*The increase in variability at these altitudes is apparent in subsequent figures.*

In page 7 line 22 (or more easily in Figure 5), the optimized criteria is chosen for geophysical variability is less than 10%. However according to page 2 line 6 "Collocated measurements should be close to each other relative to the spatiotemporal scale on which the variability of the geophysical field becomes comparable to the measurement uncertainties", shouldn't then the optimized criteria be for less than the combined measurement uncertainties. According to section 2.1, "on the order of a few percent for ACE-FTS" (please be more specific) and according to section 2.2 within 5% for OSIRIS. Is the combined measurement uncertainty less than 10%, is using the combined measurement uncertainty as criteria to strict? so that only a few coincidences are found? What are the implications of this?

*We weren't trying to say that 10% is the optimal level of geophysical variability. The text is now clearer that the point is that with this technique, one can choose a level of variability (we happened to choose 10%) and optimize the coincidence criteria accordingly.*

Also, please add in figure 5 the difference in percent so that it is easily comparable with the rest of the figures.

*The left panel is in ppmv, and the right panel is in percent. Maybe the confusion is that the x-axis in the left panel had a missing "$2\sigma$"? The $2\sigma$ has been added.*

In page 7 line 24-26, the manuscript will be enriched showing an example of the biases that can be induced by having different coincidence criteria per height. Please quantify it.

*We don't believe this would be useful, as it is simply a warning that if this technique is used, one must ensure that there aren't drastic changes in season/latitude with altitude due to changing criteria. (The magnitude will be different for different instruments and target species).*

Page 8 line 14: after "It is also interesting to observe the difference in geophysical variability between the polar NH (poleward of 50◦N) region and the polar SH (poleward of 50◦S) region" please add: where most of the ozone variability can be found.

*We added "where there is greater $O_3$ variability in general."*

I suggest splitting the analysis of Figure 8 into two periods, during polar vortex season and during the offseason. That way, the sentences about stronger descent, SSWs, and in or out the vortex will be more certain. And it will presumably showcase that the coincidence criteria to maintain a given geophysical variability criteria will vary with season.

*Thank you for the suggestion. Because of this, we found that there were significant sampling biases by not taking season into account! Figure 8 has been split into different months for both hemispheres. This is also now discussed in the text, "Above 15 km in the summer months, when there is not a strong polar vortex, the NH and SH exhibit similar geophysical variability profiles, with variability on the order of 5-15%. In the same altitude region in the SH spring, geophysical variability is much larger, due to the strong and prevalent Southern polar vortex, which is just starting to break up with the onset of sunlight; and at laxer coincidence criteria, it is more likely that one instrument will be observing inside the Southern polar vortex and the other outside the vortex, which can have different atmospheric conditions. The variability is on the order of 15-20% above 22 km, and peaks at 35% near 18 km, where there is some of the most ozone depletion. As can be seen on the left panel of Fig. 7, in the lower stratosphere in the polar SH, the geophysical variability is more sensitive to the time coincidence criterion than in the polar NH. The NH geophysical variability above 30 km is also greater at the end of winter (~5-10%) than during the summer (~10-15%). This could be due to stronger planetary wave forcing in the NH (e.g. Butchart, 2014; de la Cámara, 2018) and/or stronger descent of NO and $NO_2$ following sudden stratospheric warming events (e.g. Reddmann et al., 2010)."*

Also, consider exploring the tropics separately where the tropopause is higher implying difference criteria for lower (tropospheric) altitudes.

*Due to ACE orbit, the majority of colocations is in the extra-tropics, making it much more difficult to focus on the tropics.*

Please add the mean of the relative differences panel to figure 9. So that Figure 6 and Figure 9 have the same layout.

*Bias profiles have been added to Figure 9 so that it is consistent with Figure 6.*

Summary or all the manuscript really: Please make clear that the optimize criteria discussed are only valid for the ACE-FTS and OSIRIS pairings used in this study. That is, that a similar analysis will have to be conducted for all pair of instruments to be compared.

*This has been made clearer in the introduction, "It is important to note that this study is not intended to validate either the ACE-FTS or OSIRIS O₃ data products. This is a case study that makes use of ACE-FTS and OSIRIS geolocation data and O₃ products to demonstrate how readily available data from nudged climate models can be used to estimate large scale geophysical variability between satellite measurements of atmospheric trace species, and how they can be used to make informed decisions when choosing coincidence criteria in a validation study."*

*And in the summary, "This technique of using the natural variability estimates in order to optimize the coincidence criteria can however also be used for data that is isolated to a single season/latitude range."*

**Referee #2**

MAIN COMMENTS

1) The authors characterize the natural variability of the ozone field using the data from 3 CCM/CTM models having rather low spatial resolution, from 1.9 deg to 3.75deg. This implies that the lowest spatial scale that can be probed with these models is ~200-300 km. The smaller-scale variability is not resolved by the models and cannot be characterized. This should be at least stated clearly in the paper (in particular, p.2 line 29, "large scale" should be quantified). However, the optimal way would be the inclusion of simulations with a high-resolution CTM.

*We now characterize "large scale" in the text, "In this study, given the horizontal resolution of the three climate models that were used, large scale variability is on the order of 200-300 km, which is on the order of the atmospheric path length of a limb viewing instrument at the tangent height."*

2) The characterization of ozone variability assessed using the model data is too simplified – both in general and for the considered application of satellite data validation. First, the variability ozone variability depends on latitude and season (in addition to diurnal variability). Second, the variability is not isotropic in latitude-longitude direction therefore a simple characterization by "separation distance" is too superficial. Using the model data, the spatio-temporal variability of

ozone field can be characterized in more detail, and thus the collocation criteria can be used in more advanced way (see also below).

*It is not uncommon for validation between satellite instruments to be done on a global scale, which is the case study we've chosen here. Clearly, this same technique could be used to determine the average natural variability between two datasets that are more constricted by season and latitude. This is now discussed in the text, "The coincidence criteria can be optimized for any chosen limit of geophysical variability (10% was chosen in this case), and naturally this could be done for any subset of seasons/latitudes within the collocated data."*

3) The idea of collocation criterion using the information about the natural variability obtained from modelling is good. It should be described in more detail how technically the collocation criterion "variability < 10%" is applied. Do I understand correctly that you use time-space collocation criteria as shown by circles in Figure 4, i.e., these are globally for each altitude level?

My main concern is that 10 % threshold is not actually optimized. Why have you selected 10% as a threshold? It seems to be significantly larger than uncertainties of each satellite dataset, thus the objective stated in the introduction, "Collocated measurements should be close to each other relative to the spatiotemporal scale on which the variability of the geophysical field becomes comparable to the measurement uncertainties" is not satisfied. On other hand, in the tropical middle stratosphere, for example, the overall variability is ∼5%, thus the criterion <10% variability will be satisfied automatically for any collocation criteria.

I think that the maximization of number of collocations within a variability window is not the best approach, since selecting a broader spatio-temporal window increase both number of collocations and natural variability. Instead, reduction (or minimization) of uncertainty of the bias estimates (which depends on measurements uncertainties, natural variability, and number of collocations) would be a more concrete objective, and the advantages of "optimized" criteria can be quantified (for example, reduction of bias uncertainty from x% to y%).

*We are not claiming that applying 10% natural variability globally at each altitude level is optimal. We are demonstrating that for whatever threshold of variation is desired (in this case we arbitrarily chose 10%) you can use this technique to optimize how many profiles you are using. Yes, if we were doing a validation study only comparing profiles in tropical middle stratosphere, we would likely want to cap our natural variability at a lower value than 10%. E.g., we could cap it at 3% and maximize the number of profiles used within that variability window. This is now made clearer in the text, see above response.*

Since the ozone variability strongly depends on location/season, it is expected that the optimized collocation criteria will also depend on location and season, or, at least, characterized into "low" and "high" ozone variability. At the same time, this would be reduce the drawback that you mentioned in the paper on page 7: " One drawback to having different coincidence criteria at each altitude is that it can potentially add biases between altitudes due to changing seasonal and latitudinal sampling".

*That is a good point, there is nothing stopping anyone from using this technique for non-global comparisons. The same technique could be used in separate location/seasonal bins, and it would show that the same coincidence criteria for different "bins" is not only not necessary but would lead to different influences due to natural variability and is therefore not desirable. This is now discussed in the text, "The coincidence criteria can be optimized for any chosen limit of geophysical variability (10% was chosen in this case), and naturally this could be done for any subset of seasons/latitudes within the collocated data. Although, one drawback to having different coincidence criteria at each altitude, especially when making global comparisons, is that it can potentially add biases between altitudes due to changing seasonal and latitudinal sampling. Therefore, care must be taken to ensure that biases of this type are not being introduced."*

DETAILED COMMENTS

1) P.2 , l. 29: please quantify "large scale" term

*This is now quantified, as discussed above.*

2) Section 2. Please add estimates of random uncertainties of ACE-FTS and OSIRIS ozone profiles.

*A brief discussion of the reported OSIRIS and ACE-FTS uncertainties (which are not necessarily purely random) are now included.*

*"The reported statistical fitting error, described by Boone et al. (2005; 2013), is typically on the order of 2-3% in the 10-15 km range and ~1.5-2% in the 15-55 km range."*

*"The reported OSIRIS $O_3$ uncertainties are typically on the order of 3-9% in the 10-55 km range."*

3) Section 2.2. Why don't you use version 5.10, which, as you explained in the paper, is better than v.5.07?

*When we first started the paper we only had v5.07. Since this study is not meant to focus on the validation of the individual satellite data sets (the data sets are simply examples of measured differences), the analysis has not been updated to v5.10, but tests have been done on v5.10 and the differences are typically within 1-2%. This does not change the conclusions of the study.*

4) P.5, lines 17-19: You use rather relaxed collocation criteria (12 h and 2000 km); what is the difference in pressure profiles for large separations and how this affects transformation of OSIRIS data to pressure grid? Is the pressure-altitude conversion using reanalysis data at OSIRIS locations less accurate?

*We expect that ACE pressures to be somewhat more reliable at the upper altitudes, but not at the lower altitudes. We've done a sensitivity study that shows the difference between using the ACE pressures vs the reanalysis pressures (see response to reviewer #1), which show that there isn't a significant impact to the conclusions.*

5) Why do you define sigma_nat as 2* std (MOD_ace-MOD_OSIRIS)? One would expect sigma_nat= std (MOD_ace-MOD_OSIRIS)/sqrt(2).

*Unless we're missing something obvious (which is quite possible), is the sqrt(2) coming from assuming the natural variation would be calculated as the standard error of the mean? We don't believe that would be the case. We would use the standard error of the mean if we thought there should be no difference in concentrations between the two locations and therefore trying to determine the uncertainty. But we know that separate locations have different values and we're trying to define the full range of expected differences between the measurements.*

*We now realize that labelling the geophysical variability as $\sigma_{nat}$ could be problematic, as it could be interpreted as 1 standard deviation, so we now give geophysical variability the symbol $v_{geo}$. Also, we've replaced all instances of "$2\sigma$ geophysical variability" with "$2\sigma$ variability".*

6) I suggest revision of Section 4, according to the MAIN COMMENTS 2 and 3. The spatio-temporal ozone variability (time, altitude, latitude, longitude, season) can be in detail characterized using the model data. For optimization based on variability, I suggest a categorization at least of "low" and "high" ozone variability (alternatively, according to latitude zone and season). I suggest also quantitative estimates of validation improvement (for example, reduction of uncertainties of bias estimate, bias detectability, quality of the spread estimate) based on the optimized collocation criteria.

*We now discuss that this technique can be used in different, more specific seasons/locations and is not restricted to global comparisons, see above responses.*

7) Section 4.2: The ozone variability in polar regions depends strongly on season. This should be taken into account in the analyses.

*Thank you for the suggestion. Because of this, we found that there were significant sampling biases by not taking season into account! Figure 8 has been split into different months for both hemispheres. This is also now discussed in the text, "Above 15 km in the summer months, when there is not a strong polar vortex, the NH and SH exhibit similar geophysical variability profiles, with variability on the order of 5-15%. In the same altitude region in the SH spring, geophysical variability is much larger, due to the strong and prevalent Southern polar vortex, which is just starting to break up with the onset of sunlight; and at laxer coincidence criteria, it is more likely that one instrument will be observing inside the Southern polar vortex and the other outside the vortex, which can have different atmospheric conditions. The variability is on the order of 15-20% above 22 km, and peaks at 35% near 18 km, where there is some of the most ozone depletion. As can be seen on the left panel of Fig. 7, in the lower stratosphere in the polar SH, the geophysical variability is more sensitive to the time coincidence criterion than in the polar NH. The NH geophysical variability above 30 km is also greater at the end of winter (~5-10%) than during the summer (~10-15%). This could be due to stronger planetary wave forcing in the NH (e.g. Butchart, 2014; de la Cámara, 2018) and/or stronger descent of NO and $NO_2$ following sudden stratospheric warming events (e.g. Reddmann et al., 2010)."*

---

## Referee Report (RR1)

**Review of Model estimations of geophysical variability between satellite measurements of ozone profiles by Patrick Sheese et al**

This paper discusses the geophysical variability encounter in the comparison of ACE-FTS and OSIRS for different coincident criteria using model fields. The author addressed successfully many comments in the new version, only a few minor comments remain.

Minor comments:

P6 line 22: The discussion of Figure 1 uses ppbv as well as ppmv. Please use only ppmv to improve readability.

Figure 5 x-label should be deltaVMR [%] not 2sigma deltaVMR [%], the figure shows the bias and the 2sigma geophysical variability.

Figure 8, Please include the bias (the layout can be similar to figure 2).

Figure 9 was updated but the caption was not (i.e., there is no description for the first panel)

---

## Author Response (AR2)

**Response to Reviewers - Model estimations of geophysical variability between satellite measurements of ozone profiles" by Patrick E. Sheese et al.**

*We'd like to thank reviewer #1 for their comments. Here we address minor comments, with their comments in black and our responses in green. We also note where minor corrections were made in the manuscript.*

**Referee #1**

P6 line 22: The discussion of Figure 1 uses ppbv as well as ppmv. Please use only ppmv to improve readability.

This has been updated in the text as: *"All three models exhibit a small bias (within 0.02 ppmv, 0.5%) between 12 and 29 km. Between 30 and 45 km, the model results indicate that ACE-FTS $O_3$ values are expected to be systematically lower than OSIRIS. CMAM indicates a bias of up to ~0.02 ppmv (0.5%) in this region, EMAC indicates a bias of up to ~0.06 ppmv (1.1%), and up to 0.13 ppmv (2.8%) with WACCM. Above 48 km, all three models exhibit systematically larger concentrations of ACE-FTS $O_3$ than OSIRIS $O_3$. EMAC indicates a bias of up to ~0.03 ppmv (2.1%) in this region, CMAM indicates a bias of up to 0.05 ppmv (3.9%), and up to 0.10 ppmv (8.7%) with WACCM."*

Figure 5 x-label should be deltaVMR [%] not 2sigma deltaVMR [%], the figure shows the bias and the 2sigma geophysical variability.

This has been corrected.

Figure 8, Please include the bias (the layout can be similar to figure 2).

The panel showing the bias has been added and the caption updated. This is also now discussed in the text (Page 9, line 3): *"Figure 8 shows the difference in ensemble mean bias and 2σ geophysical variability at coincidence criteria of 8 h, 1000 km."*

Figure 9 was updated but the caption was not (i.e., there is no description for the first panel)

The caption has been updated.

**Additional Corrections made by authors:**

Page 9, lines 1-2: The values in this sentence have been updated.

Figure 2: The x-axis label on the left hand-plot has been corrected.

[revised manuscript text omitted]